# Interstitial Atom Engineering in Magnetic Materials

**Jiro Kitagawa [1],\*** , **Kohei Sakaguchi [1]**, **Tomohiro Hara [1]**, **Fumiaki Hirano [1]**, **Naoki Shirakawa [2]** and **Masami Tsubota [3]**

[1] Department of Electrical Engineering, Faculty of Engineering, Fukuoka Institute of Technology, 3-30-1 Wajiro-higashi, Higashi-ku, Fukuoka 811-0295, Japan; s1451020@bene.fit.ac.jp (K.S.); s1552037@bene.fit.ac.jp (T.H.); s1652044@bene.fit.ac.jp (F.H.)

[2] Flexible Electronics Research Center, National Institute of Advanced Industrial Science and Technology, Tsukuba, Ibaraki 305-8565, Japan; shirakawa.n@aist.go.jp

[3] Physonit Inc., 6-10 Minami-Horikawa, Kaita Aki, Hiroshima 736-0044, Japan; tsubota@physonit.jp

\* Correspondence: j-kitagawa@fit.ac.jp; Tel.: +81-(0)92-606-4579

**Abstract:** Interstitial light elements play an important role in magnetic materials by improving the magnetic properties through changes of the unit cell volume or through orbital hybridization between the magnetic and interstitial atoms. In this review focusing on the effects of interstitial atoms in Mn-based compounds, which are not well researched, the studies of interstitial atoms in three kinds of magnetic materials (rare-earth Fe-, Mn-, and rare-earth-based compounds) are surveyed. The prominent features of Mn-based compounds are interstitial-atom-induced changes or additional formation of magnetism—either a change from antiferromagnetism (paramagnetism) to ferromagnetism or an additional formation of ferromagnetism. It is noted that in some cases, ferromagnetic coupling can be abruptly caused by a small number of interstitial atoms, which has been overlooked in previous research on rare-earth Fe-based compounds. We also present candidates of Mn compounds, which enable changes of the magnetic state. The Mn-based compounds are particularly important for the easy fabrication of highly functional magnetic devices, as they allow on-demand control of magnetism without causing a large lattice mismatch, among other advantages.

**Keywords:** interstitial atom; Mn-based compounds; rare-earth Fe-based compounds; rare-earth-based compound; Bethe–Slater curve; ferromagnetic; antiferromagnetic; permanent magnet

## 1. Introduction

Some crystal structures possess interstitial crystallographic sites, which light elements such as hydrogen, boron, carbon, nitrogen, and oxygen atoms can occupy. There is a rather long history of metallurgical, physical, and chemical research studies on interstitial atoms [1,2]. In past years, domain control of ferromagnets has been studied [1]. Since the 1980s, interstitial atoms have attracted intense attention related to the improvement of magnetic properties of rare-earth Fe and Co-based permanent magnets [2–6].

Interstitial atoms have two major roles—influencing the stability of the crystal structure and in the modification or change in magnetic properties. In the former case, small amounts of interstitial light elements are required to stabilize the desired crystal structure; compounds without interstitial atoms would not exist in thermal equilibrium. In the latter case, interstitial atoms affect the crystal structure parameters, such as the interatomic distances between magnetic atoms or the orbital hybridization between magnetic and interstitial atoms, consequently meaning the magnetic ordering temperature, the magnetic moment, the magnetic structure, and other factors can be altered.

The most well-studied platforms for interstitial atoms are rare-earth Fe-based permanent magnets. The improvements of the magnetic properties have mainly been achieved through the addition of light

elements such as boron, carbon, and nitrogen atoms. The Bethe–Slater curve is one of the criteria needed to understand whether metal $3d$ transition elements of Cr, Mn, Fe, Co, and Ni possess ferromagnetic (FM) or antiferromagnetic (AFM) states (see Figure 1) [7–10]. This curve exhibits the exchange coupling as a function of the interatomic distance. Fe falls in the FM region near to the border between FM and AFM states. Therefore, in Fe-based compounds, a shorter Fe–Fe distance (shrinkage of the unit cell volume) favors an AFM state, while a longer Fe–Fe distance (an expansion of the unit cell volume) favors the FM state [11–13]. With increasing Fe–Fe distance, smaller overlapping of $3d$ wave functions makes the $3d$ band narrower, which leads to the FM state, and in most cases the Curie temperature $T_C$ is enhanced.

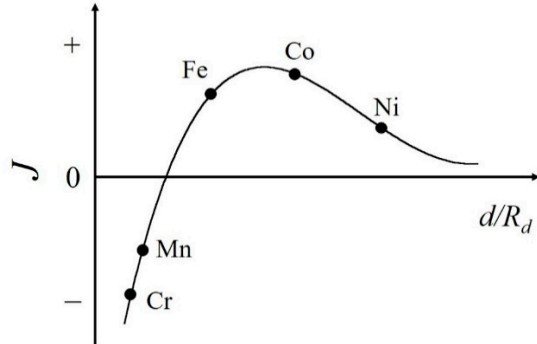

**Figure 1.** Schematic view of the Bethe–Slater curve. Here, $J$, $d$, and $R_d$ represent the magnetic exchange coupling between atoms, the interatomic distance, and the radius of the $3d$ shell, respectively.

On the other hand, the effects of interstitial atoms in Mn-based compounds are not well researched. As shown in Figure 1, the Mn atom itself shows the AFM ground state, however the expanded Mn–Mn distance in Mn-based compounds leads to the FM state. Mn compounds are indispensable for both FM and AFM materials. For example, MnBi and MnAl have attracted much attention as permanent magnets [14–18]. MnSi has been extensively studied as a magnetic material with a skyrmion state, which is a noncollinear magnetic structure. The skyrmion domains can be driven by the low current density threshold [19]. Recently, $Mn_3Sn$ has been intensively studied as a topological antiferromagnet [20] and is a candidate next-generation spintronics material. If the effects of interstitial atoms are well understood, they can be highly useful in the development of spintronics devices or highly functional magnetic devices, which can be developed via easy on-demand control of the magnetic state.

In this review, we present information on the effects of interstitial atoms in rare-earth Fe-, Mn-, and rare-earth-based compounds, especially focusing on Mn-based compounds. We introduce the frequently studied Mn-based compounds, showing the changes of magnetic states caused by the interstitial atoms, as well as our recent results. Interestingly, a change in the magnetic ground state or an additional formation of the FM state caused by interstitial atoms is possible. Specifically, our results highlight an abrupt emergence of FM exchange coupling above room temperature caused by the addition of a small amount of light elements, which has not been clearly reported for rare-earth Fe-based compounds. Additionally, candidate crystal structures for Mn compounds are presented, in which a magnetic state can be changed using interstitial atoms. Finally, perspectives concerning the development of magnetic devices based on Mn-based compounds and the strategy for further improvement of magnetic properties are outlined.

## 2. Rare-Earth Fe-Based Compounds

### 2.1. $Th_2Zn_{17}$-Type

$R_2Fe_{17}N_x$ (R = light rare-earth) represents the permanent magnets with the hexagonal $Th_2Zn_{17}$-type structure (space group: $R\bar{3}m$, No. 166). There is an interstitial Wyckoff position $9e$ for nitrogen

atoms. The roles of nitrogen atoms are the enhancement of $T_C$ and the saturated magnetization $M_s$ and the change in magnetic anisotropy. For example, $T_C = 389$ K and $M_s = 1.00$ T at a room temperature of $Sm_2Fe_{17}$ are increased to 749 K and 1.54 T, respectively, in $Sm_2Fe_{17}N_3$ (see Table 1). Moreover, the in-plane anisotropy in $Sm_2Fe_{17}$ is changed to uniaxial anisotropy by the nitriding, which is advantageous for application as permanent magnets.

**Table 1.** The magnetic and structural properties of rare-earth Fe-based compounds with and without interstitial atoms, where *a* and *c* are the lattice parameters, and *V* is the unit cell volume. RT means the room temperature. Data are from the references listed in the table.

| Compound | $T_C$ (K) | $M_S$ @RT (T) | Easy Magnetization Direction | *a* (Å) | *c* (Å) | *V* (Å³) | Ref. |
|---|---|---|---|---|---|---|---|
| $Sm_2Fe_{17}$ | 389 | 1.00 | in-plane | 8.55 | 12.45 | 788.2 | [21,22] |
| $Sm_2Fe_{17}N_3$ | 749 | 1.54 | c axis | 8.74 | 12.70 | 840.2 | [23,24] |
| $Sm_2Fe_{17}C_3$ | 668 | 1.43 | c axis | 8.744 | 12.572 | 832.4 | [25,26] |
| $SmFe_{11}Ti$ | 584 | 1.15 | c axis | 8.56 | 4.80 | 351.7 | [27,28] |
| $SmFe_{11}TiN$ | 769 | 1.28 | in-plane | 8.64 | 4.84 | 361.3 | [28,29] |
| $NdFe_{11}Ti$ | 547 | 1.38 | in-plane | 8.585 | 4.789 | 353.0 | [30,31] |
| $NdFe_{11}TiN$ | 729 | 1.48 | c axis | 8.701 | 4.844 | 366.7 | [32,33] |
| $SmFe_9Si_2C_{1.0}$ | 462 | 0.88 | c axis | 10.057 | 6.512 | 658.7 | [34] |
| $SmFe_9Si_2C_{1.5}$ | 492 | 0.90 | c axis | 10.082 | 6.554 | 666.1 | [34] |

The addition of nitrogen atoms expands the lattice parameters *a* and *c*, resulting in the increased Fe–Fe interatomic distance and the enhancement of $T_C$. Actually, in $Sm_2Fe_{17}N_3$, $T_C$ is almost doubled, which is much larger than that in the case of $ThMn_{12}$-type compounds (see also Section 2.2 and Table 1). The difference is well correlated with the rate of unit-cell-volume change $\Delta V/V$ by the interstitial atoms, which is defined by

$$\frac{\Delta V}{V} = \frac{V_w - V_{w/o}}{V_{w/o}} \times 100 \tag{1}$$

where $V_{w/o}$ and $V_w$ are the unit cell volumes without and with interstitial atoms, respectively. We note here that the compound without interstitial atoms is hereafter often called the parent compound. For example, $\Delta V/V = 6.6\%$ in $Sm_2Fe_{17}N_3$ is larger than that in $ThMn_{12}$-type $SmFe_{11}TiN$ (2.7%). The larger $\Delta V/V$ for $Sm_2Fe_{17}N_3$ is ascribed to the fact that Fe magnetic moments in the $Th_2Zn_{17}$-type structure are less itinerant.

## 2.2. ThMn₁₂-Type

The well studied system is $RFe_{11}Ti$ (R: rare-earth) series (*I4/mmm*, No. 139). This structure allows the interstitial nitrogen atoms to occupy the 2*b* site. In $SmFe_{11}TiN$ or $NdFe_{11}TiN$, the nitrogen addition enhances $T_C$ by about 30%, which is smaller than that of the $Th_2Zn_{17}$-type system (see Table 1). As mentioned above this is due to the smaller $\Delta V/V$ under the addition of nitrogen (2.7% for $SmFe_{11}TiN$ and 3.9% for $NdFe_{11}TiN$). The magnetic anisotropy of $RFe_{11}Ti$ is determined primarily due to the single-ion contribution of R ion, thus $NdFe_{11}Ti$ and $SmFe_{11}Ti$ show the planar and *c* axis anisotropies, respectively. In each compound, the sign of magnetic anisotropy constant $K_1$ is reversed by the nitrogen addition and a good candidate of the permanent magnet is consequently $NdFe_{11}TiN$.

The first-principle studies have revealed that $NdFe_{12}N$, which possesses an Fe concentration higher than that of $NdFe_{11}TiN$ and is more favorable for a permanent magnet, has excellent magnetic properties [35–37], however, it is thermodynamically unstable in bulk form. The thin film fabrication allows the growth of $NdFe_{12}N_x$, which shows a superior $M_s = 1.7$ T compared to Nd–Fe–B magnets [38]. The coercivity of $NdFe_{12}N_x$ is rather low, however, a born doped $Sm(Fe_{0.8}Co_{0.2})_{12}$ is recently reported to be a highly coercive material with 1.2 T [39]. The microstructure of the compound exhibits columnar-shaped $Sm(Fe_{0.8}Co_{0.2})_{12}$ grains, surrounded by a born-rich amorphous phase. The domain wall pinning at the grain boundary is responsible for the high coercivity [39].

Mao et al. reported a *BH* energy product of ThMn$_{12}$-type compound [40]. They have investigated the magnetic properties of powdered PrFe$_{12-x}$V$_x$ and the nitride compound. Pr(Fe, V)$_{12}$N$_{1.6}$ exhibits the maximum *BH* energy product of 135 kJ/m$^3$.

### 2.3. BaCd$_{11}$-Type

The tetragonal BaCd$_{11}$-type structure represented by RFe$_9$Si$_2$C$_x$ (Table 1) is a candidate for a next-generation rare-earth Fe-based permanent magnet. It is interesting that the addition of carbon atoms is required to stabilize the BaCd$_{11}$-type structure. In RFe$_9$Si$_2$C$_x$, the carbon atom occupies the interstitial 8*c* site of the tetragonal space group of *I*4$_1$/*amd* (No. 141). $T_C$ progressively increases with the increase of x, for example, $T_C$ = 367 K of SmFe$_9$Si$_2$C$_{0.5}$ is enhanced to 492 K in SmFe$_9$Si$_2$C$_{1.5}$, accompanying the volume expansion [30]. The magnetic anisotropy depends on the R atom; in-plane anisotropy is observed in R = Ce or Nd, whilst on the other hand, R = Sm possesses the *c* axis anisotropy.

### 2.4. Remark on Hybridization Effect

Through the extensive studies of rare-earth Fe-based permanent magnets, it is revealed that stronger orbital hybridization in the covalent bond between Fe and an interstitial atom tends to reduce the magnetic moment. The strength of the covalent bond is well reflected by $\Delta V/V$ under the addition of light elements; $\Delta V/V$ is not so large when the hybridization is stronger. $\Delta V/V$ is generally larger for the nitrogen addition compared to the carbon one (see, for example, Sm$_2$Fe$_{17}$N$_3$ and Sm$_2$Fe$_{17}$C$_3$ in Table 1).

## 3. Mn-Based Compounds

In this section, we gathered as many Mn-based room temperature ferromagnets as possible.

### 3.1. Hydrogen-Absorbed (R or Th)$_6$Mn$_{23}$

R$_6$Mn$_{23}$ (R = rare earth) series and Th$_6$Mn$_{23}$ crystallize into the cubic Th$_6$Mn$_{23}$-type structure with the space group $Fm\bar{3}m$ (No. 225). R atoms occupy the 24*e* site, and Mn atoms the 4*b*, the 24*d* and the two 32*f* sites. All R$_6$Mn$_{23}$ compounds show FM orderings at a $T_C$ higher than 300 K, which are induced by Mn magnetic moments. On the other hand, Th$_6$Mn$_{23}$ remains a paramagnet down to low temperatures. The hydrogen absorption generally expands the unit cell volume (see Figure 2 and Table 4), whereas the occupation site of the hydrogen atom is not clear. Although the $T_C$ of non-hydrogenated R$_6$Mn$_{23}$ rises with increasing volume, most of the hydrogen-absorbed compound shows the suppressed $T_C$ compared to each parent compound. For example [41], $T_C$ = 461 K of Gd$_6$Mn$_{23}$ is reduced to $T_C$ = 2.66 K in Gd$_6$Mn$_{23}$H$_x$. The saturated moment is also highly reduced (49 μ$_B$/f.u. to 14.2 μ$_B$/f.u.) [41]. The opposite behavior is confirmed for Th$_6$Mn$_{23}$, in spite of the similar volume change by the hydrogenation. The hydrogenated Th$_6$Mn$_{23}$ shows an emergence of ferromagnetism with $T_C$ = 335 K, although the saturated moment of Mn is not so high (16.5 μ$_B$/f.u.) [41]. Considering that R and Th ions are in the trivalent and the tetravalent state, respectively, the exchange coupling between Mn magnetic moments would significantly depend on the valence electron count per atom (VEC). We note that Th$_6$Mn$_{23}$ is the typical Mn-based compound showing the change in magnetic state by interstitial atoms. The saturated moment induced in the FM state of the hydrogenated compound is rather low, indicating a strong orbital hybridization between Mn and hydrogen atoms and/or a complex magnetic structure.

### 3.2. Hydrogen-Absorbed YMn$_2$

The crystal structure of YMn$_2$ is the cubic MgCu$_2$-type structure with the space group of $Fd\bar{3}m$ (No. 227). Y and Mn atoms occupy the 8*a* and the 16*d* site, respectively. YMn$_2$ can absorb hydrogen atoms as in R$_6$Mn$_{23}$. Although YMn$_2$ is paramagnetic down to low temperatures, YMn$_2$H$_x$ shows a lattice expansion under hydrogenation, which induces a ferromagnetism with $T_C$ = 284 K and a

saturation moment of 0.52 $\mu_B$/f.u. [41]. There are many $RMn_2$ compounds with the same structure, however, we do not make the magnetic ordering temperature vs. unit cell volume plot, because the accurate nature of the magnetic order is still unknown for each compound [42]. $YMn_2$ can also be regarded as the compound, realizing the change from paramagnetism to ferromagnetism by interstitial atoms.

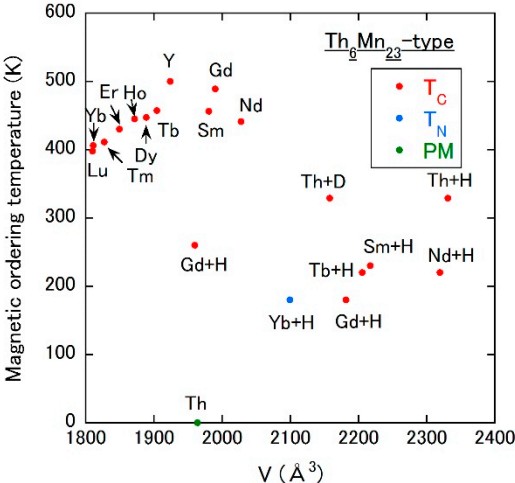

**Figure 2.** Magnetic ordering temperature vs. unit cell volume plot of $Th_6Mn_{23}$-type Mn compounds. $T_N$ is the Néel temperature. PM means the paramagnetic state. For example, Th + H (Th + D) means the hydrogen (deuterium)-absorbed $Th_6Mn_{23}$.

### 3.3. Carbon-Added $Mn_5Si_3$

This system shows a change from the AFM to FM state by the carbon addition. The hexagonal $Mn_5Si_3$-type structure is known as a superior platform for studying interstitial chemistry [43]. The space group is $P6_3/mcm$ (No. 193), in which the Wyckoff positions are the $4d$ for Mn1, the $6g$ for Mn2 and the $6g$ for Si (see also Figure 3a). Carbon atoms occupy the $2b$ site. While the parent compound $Mn_5Si_3$ shows an AFM ordering at the Néel temperature $T_N$ = 98 K, a thin film $Mn_5Si_3C_x$ becomes a room-temperature ferromagnet with $T_C$ = 350 K as shown in Figure 3b (see the filled squares) [44,45]. In [44], the appearance of the FM state is ascribed to the enhanced Mn–Mn interaction mediated by added carbon. Associated with the unit cell volume expansion by the carbon addition, the saturation Mn moment rapidly increases to 1 $\mu_B$/Mn at x = 0.75 (see Figure 3c). However, the x dependence of $T_C$ is peculiar: a $T_C$ plateau of 350 K in a rather wide x range, where the saturation Mn moment is steadily reduced as x is increased above 0.75. We speculate that the hybridization between Mn2 and C atoms is strong, due to the short Mn2–C distance (see Figure 3a). As x is increased, the magnetic moment of Mn2 would be decreased, but that of Mn1 under well localized state due to the weak influence of the carbon addition would be enhanced. The rather strong Mn1–Mn1 magnetic interaction would be responsible for the plateau of $T_C$, while the decreasing Mn2 moment with increasing x would contribute to the reduction in the saturation moment.

### 3.4. (R or Actinide)$Mn_2Si_2$ and Its Germanides

Despite the absence of a report on the addition of light elements in these compounds, the magnetic ordering temperature vs. unit cell volume plot shows the thought-provoking results as shown in Figure 4. These compounds possess the tetragonal $ThCr_2Si_2$-type structure (*I4/mmm*, No. 139), where Mn atoms at the $4d$ sites form the layered structure perpendicular to the *c* axis. R (Actinide) and Si (Ge) atoms occupy the $2a$ and the $4e$ sites, respectively. In the case of $RMn_2Si_2$, only the R = La compound exhibits an FM state and the other compounds AFM one (see 21–35 in Figure 4). When Si is replaced by Ge, FM (AFM) states are observed for R = La to Sm (R = the other elements) as shown in

36–47 of Figure 4. Figure 4 provides a good correlation between the magnetic ordering temperature and the unit cell volume throughout the two series: systematically descending ordering temperature with expanding volume. Furthermore, magnetic structure changes from AFM to FM at approximately 179 Å$^3$, which is consistent with the picture of the Bethe–Slater curve. The inverse trend is confirmed in (U or Th)Mn$_2$Si$_2$ and its germanide (see 48 and 49 (50 and 51) in Figure 4), that is, the crossover from the FM to AFM state occurs by increasing the volume in each system. This can be ascribed to the difference of valence between rare-earth (trivalent) and actinide (maybe tetravalent) elements as in the case of Th$_6$Mn$_{23}$-type compounds.

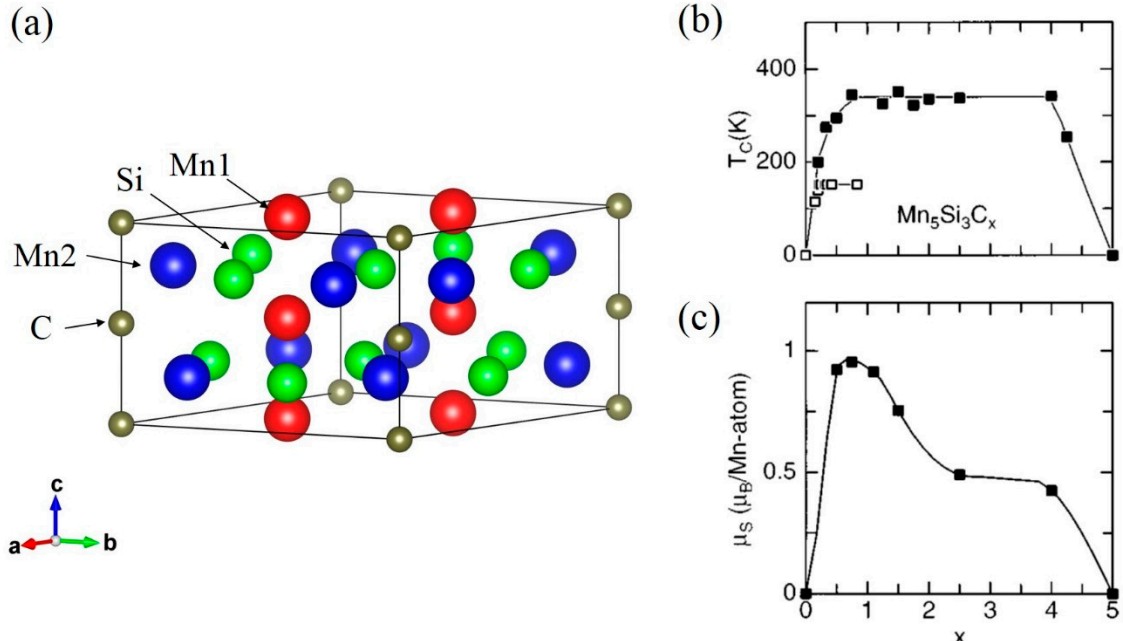

**Figure 3.** (**a**) Crystal structure of Mn$_5$Si$_3$C$_x$, where the solid line represents the unit cell. x dependence of (**b**) $T_C$ and (**c**) the saturation magnetic moment for Mn$_5$Si$_3$C$_x$. In (**b**), the filled and the open squares indicate the thin film and bulk samples, respectively. Reproduced with permission from [44].

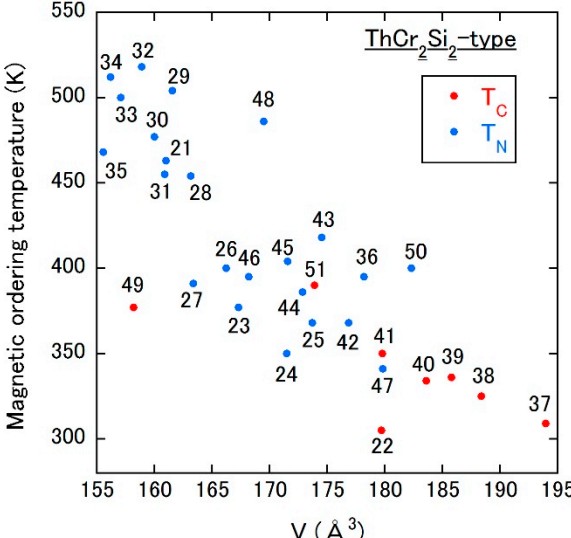

**Figure 4.** Magnetic ordering temperature vs. unit cell volume plot of ThCr$_2$Si$_2$-type Mn compounds. The numbers in the figure correspond to those in Table 4 (21–47 for R-containing compounds, and 48–51 for U- or Th-containing compounds).

### 3.5. Boron-Added Pd$_{0.75}$Mn$_{0.25}$ Alloy

The disordered face-centered-cubic (fcc) Pd$_{1-x}$Mn$_x$ alloys are the well studied spin-glass system [46]. The crystal structure is described by the space group of *Fm3̄m* (No. 225), possessing only the 4*a* site (see Figure 5), that is randomly occupied by Pd and Mn atoms. At *x* = 0.25, the spin-glass transition temperature $T_{SG}$ is 45 K [47]. The Pd$_{0.75}$Mn$_{0.25}$ alloy can incorporate boron atoms [48,49] with the solubility limit of approximately x ~ 0.16 in Pd$_{0.75}$Mn$_{0.25}$B$_x$. According to the structure report [50] of PdH$_x$ with the same structure, interstitial atoms prefer the octahedral sites (the 4*b* site, see Figure 5). The volume expansion occurs with the increase of x as shown in Table 2. Figure 6a shows the isothermal magnetization curves of Pd$_{0.75}$Mn$_{0.25}$B$_x$ at room temperature, which demonstrate the emergence of room temperature ferromagnetism by the slight addition of boron atoms. To elucidate the impact of the boron addition on the effective magnetic moment of Mn atoms, temperature dependences of inverse dc magnetization 1/$\chi_{dc}$ are measured as shown in Figure 6b. All measured samples follow the Curie–Weiss law above $T_C$ (see the solid lines in Figure 6b, and the extracted effective magnetic moment $\mu_{eff}$ and the Weiss temperature $\theta$ are summarized in Table 2). $\mu_{eff}$ of the parent compound is 4.85 $\mu_B$/Mn, which is once reduced by the boron addition, that is a signature of hybridization between the Mn and boron atoms. However, the value grows with increasing $T_C$ up to 390 K, which is the maximum value in this system. Furthermore, $\mu_{eff}$ is comparable to the saturation moment in Figure 6a. These facts suggest that the hybridization would be rather weak due to the wide space of the octahedral cavity and/or the rather low occupancy derived from being born at the octahedral site (e.g., 12.5% at x = 0.125). In the latter case, a Mn atom near the boron atom possesses a reduced magnetic moment and that with no neighboring boron atom would show an enhanced moment. Above x = 0.148, both $T_C$ and $\mu_{eff}$ are reduced, which designates the dominating orbital hybridization between Mn and boron atoms, which is also consistent with the small change in the unit cell volume.

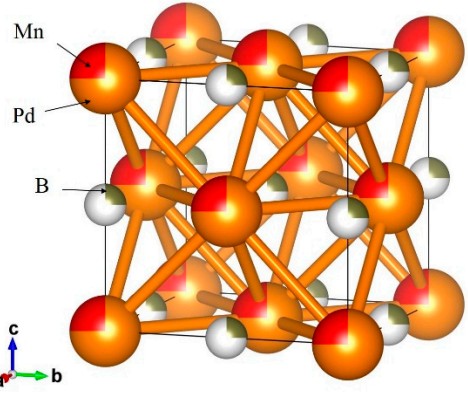

**Figure 5.** Crystal structure of Pd$_{0.75}$Mn$_{0.25}$B$_x$. The solid line represents the unit cell.

**Table 2.** Lattice parameter, unit cell volume, $\mu_{eff}$, $\theta$ and $T_C$ of Pd$_{0.75}$Mn$_{0.25}$B$_x$.

| x | *a* (Å) | *V* (Å³) | $\mu_{eff}$ ($\mu_B$/Mn) | $\theta$ (K) | $T_C$ (K) |
|---|---|---|---|---|---|
| 0 | 3.909 | 59.7 | 4.85 | −94 | - |
| 0.015 | 3.916 | 60.1 | 2.01 | 218 | 325 |
| 0.055 | 3.925 | 60.5 | 2.84 | 333 | 340 |
| 0.070 | 3.936 | 61.0 | 3.31 | 355 | 339 |
| 0.105 | 3.988 | 63.4 | 3.09 | 384 | 374 |
| 0.125 | 4.008 | 64.4 | 3.40 | 394 | 390 |
| 0.148 | 4.026 | 65.3 | 3.72 | 313 | 330 |
| 0.155 | 4.031 | 65.5 | 1.23 | 291 | 252 |
| 0.168 | 4.020 | 65.0 | 1.26 | 296 | 256 |

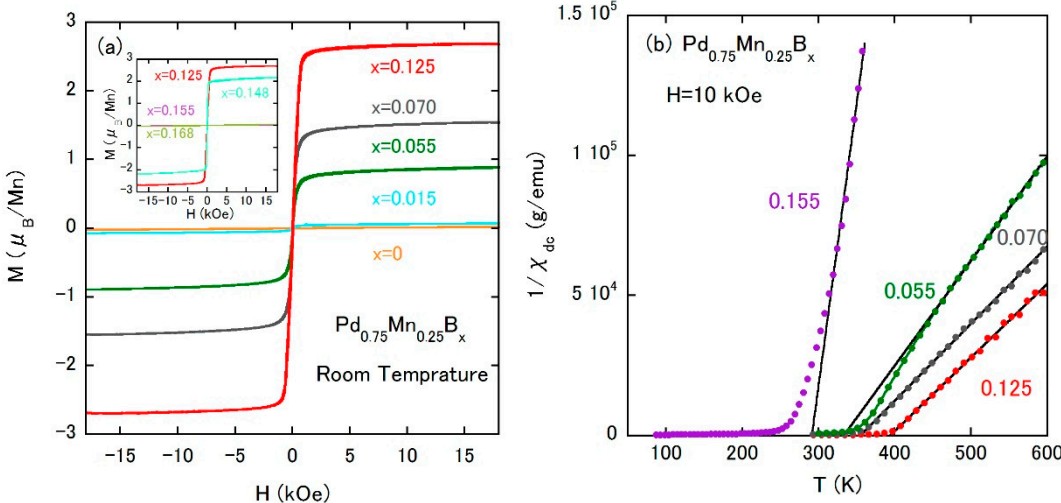

**Figure 6.** (**a**) Isothermal magnetization curves of $Pd_{0.75}Mn_{0.25}B_x$ at room temperature. Reproduced with permission from [49]; (**b**) temperature dependences of the inverse $\chi_{dc}$ of $Pd_{0.75}Mn_{0.25}B_x$. The external field is 10 kOe.

The magnetic phase diagram is constructed as shown in Figure 7a,b. At lower x, a coexistence of the fcc phase and the ordered derivative phase ($AuCu_3$) of fcc occurs [48,49]. The latter phase is responsible for another magnetic ordering at $T_M$ [49]. The spin-glass state survives under the emergence of FM state, and both ordering temperatures seem to compete with each other. Therefore, this system is considered to be the typical Mn-based compound presenting the additional formation of the FM state, coexisting with the spin-glass state of Mn atoms at low temperatures. Figure 7c displays the relationship between the $T_C$ and the unit cell volume. The vertical blue line is drawn at the volume of the parent compound showing only the spin-glass state. A linear correlation between the $T_C$ and the unit cell volume in the region of $V = 60–64 \text{ Å}^3$ indicates a $T_C$ higher than 300 K even at the blue line, which means a possible abrupt birth of FM exchange coupling.

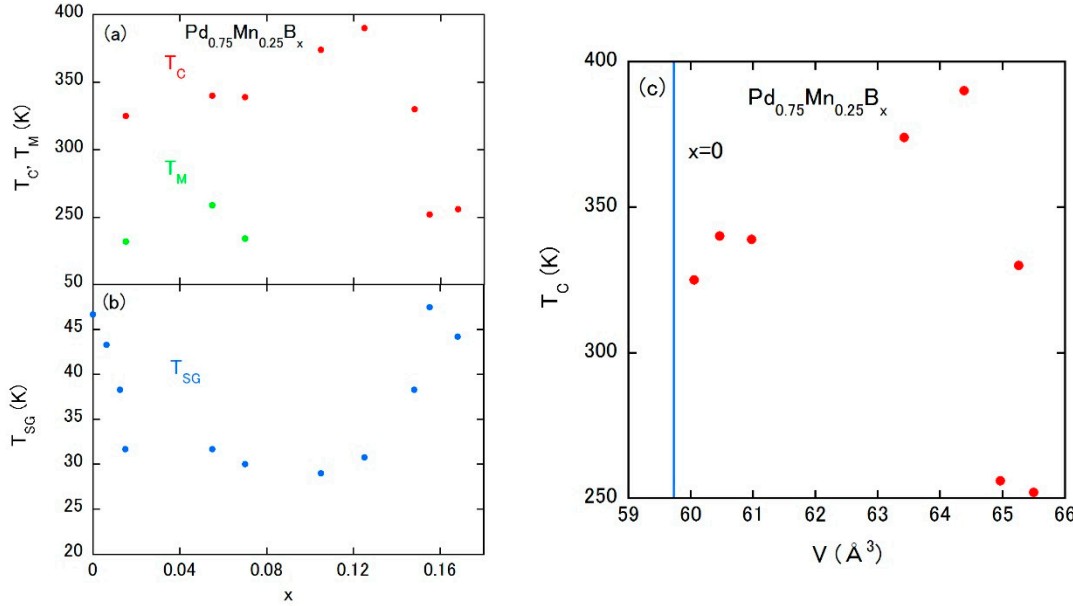

**Figure 7.** (**a**,**b**) Magnetic phase diagram of $Pd_{0.75}Mn_{0.25}B_x$; (**c**) $T_C$ vs. $V$ plot of $Pd_{0.75}Mn_{0.25}B_x$.

We remarked on the transport properties of $Pd_{0.75}Mn_{0.25}B_x$. Figure 8 shows the temperature dependences of electrical resistivity, which highlight the rather large temperature dependence

below $T_C$ even in the disordered alloy. Considering that the spin-glass parent compound shows a weak temperature dependence, the FM interaction might produce some coherence effect on the electrical conductivity.

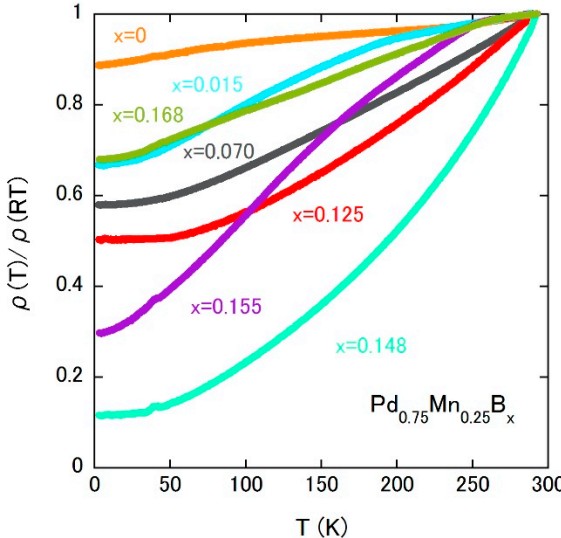

**Figure 8.** Temperature dependences of the electrical resistivity of $Pd_{0.75}Mn_{0.25}B_x$.

### 3.6. Boron-Added $Sm_2Mn_8Al_9$

$Sm_2Mn_8Al_9$ is isostructural to $R_2Fe_{17}N_x$, as mentioned in Section 2.1, and allows the interstitial B atoms at the 9*e* site (see Figure 9) [51]. In the present case, Mn and Al atoms randomly occupy the Zn sites of hexagonal $Th_2Zn_{17}$-type structure (space group: $R\bar{3}m$, No. 166). There is only one Wyckoff position 6*c* for Sm atoms, but Mn and Al atoms have four Wyckoff positions 6*c*, 9*d*, 18*f* and 18*h*, which are tentatively represented by Mn1, Mn2, Mn3 and Mn4, respectively, in Figure 9. The boron concentration dependences of lattice parameters and unit cell volume for $Sm_2Mn_8Al_9B_x$, determined with the help of the Rietveld refinement program [52], are listed in Table 3. The solubility limit would be x ~ 1. $\Delta V/V$ at x ≥ 0.1 is much smaller than that of $Pd_{0.75}Mn_{0.25}B_x$, which suggests the stronger orbital hybridization between Mn and boron atoms in $Sm_2Mn_8Al_9B_x$ (see also Section 2.4). We note here that the nearest neighbor atoms of boron are Mn3 and Mn4, which amount to 71% of all Mn atoms (see also Figure 9). This may lead to a large increase in hybridization under a small variation of volume by the interstitial atoms.

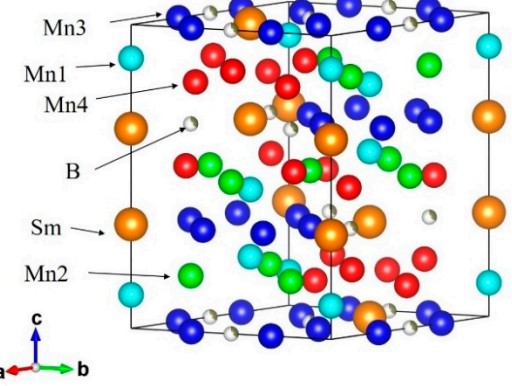

**Figure 9.** Crystal structure of $Sm_2Mn_8Al_9B_x$; the solid line represents the unit cell.

**Table 3.** Lattice parameter, unit cell volume, $\mu_{\text{eff}}$, $\theta$ and $T_C$ due to Mn moments of $Sm_2Mn_8Al_9B_x$. Assuming that a negligible contribution of the Sm magnetic moment is usually smaller than the Mn moment value, $\mu_{\text{eff}}$ is calculated.

| x | a (Å) | c (Å) | V (Å³) | $\mu_{\text{eff}}$ ($\mu_B$/Mn) | $\theta$ (K) | $T_C$ due to Mn (K) |
|---|---|---|---|---|---|---|
| 0 | 8.970 | 13.109 | 913.5 | 2.89 | −212 | - |
| 0.1 | 8.958 | 13.091 | 909.7 | 2.91 | 15 | 415(20) |
| 0.5 | 8.958 | 13.093 | 910.0 | 1.34 | 443 | 408(10) |
| 0.75 | 8.966 | 13.100 | 912.0 | 1.84 | 453 | 404(5) |
| 1 | 8.975 | 13.112 | 914.6 | 0.98 | 489 | 437 |

The isothermal magnetization curves are measured at room temperature as shown in Figure 10a, which have revealed the abrupt emergence of ferromagnetism at even a small amount of boron atoms. The temperature dependences of $1/\chi_{\text{dc}}$ demonstrate that boron-added samples follow the Curie–Weiss law above $T_C$ (see the solid lines in Figure 10b). $Sm_2Mn_8Al_9$ shows a FM behavior below approximately 100 K. Taking into account that the La-counterpart does not show a magnetic ordering at that temperature (see the ac magnetization $\chi_{\text{ac}}$ results in the inset of Figure 10b), the low-temperature magnetic transition in the parent compound is due to Sm ions. The $\mu_{\text{eff}}$ and Weiss temperature of each sample are summarized in Table 3. As x is increased, $\mu_{\text{eff}}$ is rapidly reduced, which is indicative of strong orbital hybridization, which is also supported by the weak x dependence of the unit cell volume. It is to be pointed out that the paramagnetic Mn moment abruptly forms the FM state at room temperature even when the density of boron atoms is substantially low. Under a strong orbital hybridization between Mn and boron atoms, the lattice expansion might not be essential for the formation of FM coupling at room temperature.

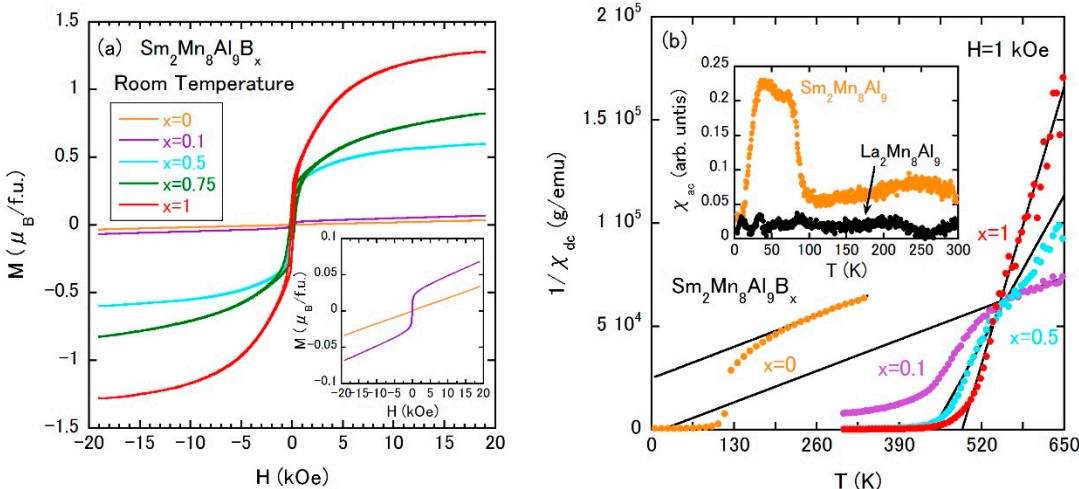

**Figure 10.** (**a**) Isothermal magnetization curves of $Sm_2Mn_8Al_9B_x$ at room temperature; (**b**) temperature dependences of the inverse $\chi_{\text{dc}}$ of $Sm_2Mn_8Al_9B_x$. The external field is 1 kOe, the inset is the temperature dependences of $\chi_{\text{ac}}$ of $Sm_2Mn_8Al_9$ and $La_2Mn_8Al_9$.

The magnetic properties of $Sm_2Mn_8Al_9B_x$ are summarized as a magnetic phase diagram in Figure 11a. The $\chi_{\text{ac}}$ (T) measurements indicate the survival of magnetic ordering due to Sm ions at approximately 85 K, which is independent of the room temperature FM state and seems to disappear at x = 1. The high-temperature $T_C$ due to Mn atoms shows a shallow minimum at x = 0.75. The inset of Figure 11a is the high-temperature $T_C$ vs. V plot, in which V at x = 0 is denoted by the vertical blue line as in Figure 7c. The observation of room temperature ferromagnetism at both sides of the vertical line strongly suggests that the birth of room temperature FM coupling between Mn atoms is

not related to the volume change. The effect of volume change, as studied well in rare-earth Fe-based magnets, is triggered by the appearance of the FM state due to Mn atoms.

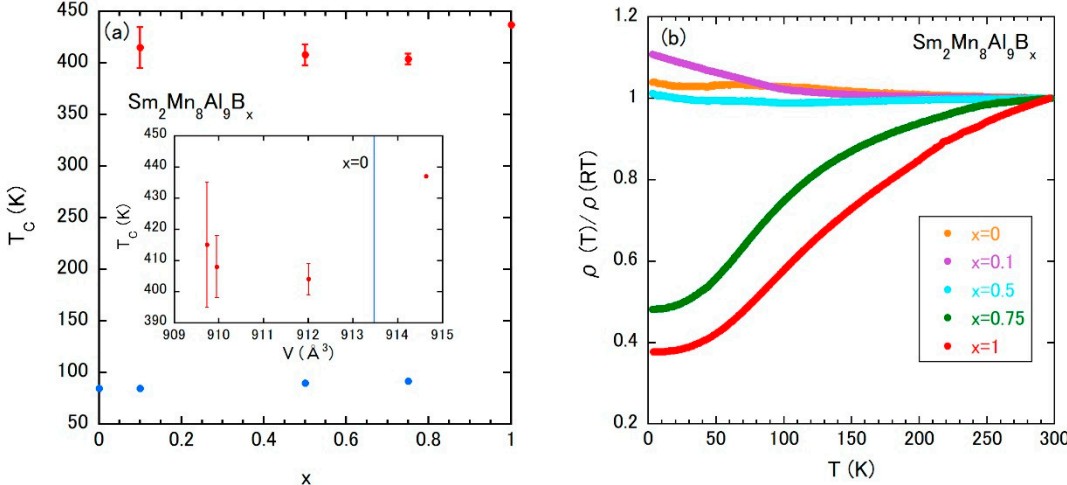

**Figure 11.** (**a**) Magnetic phase diagram of $Sm_2Mn_8Al_9B_x$; the inset is $T_C$ vs. $V$ plot of $Sm_2Mn_8Al_9B_x$; and (**b**) temperature dependences of electrical resistivity of $Sm_2Mn_8Al_9B_x$.

The temperature dependence of $\rho$ exhibits a drastic change as shown in Figure 11b. The carriers would show a localized nature up to x = 0.5, however, metallic behavior is observed above x = 0.75. The localized behavior is also reported in isostructural Mn compounds [53] such as $Gd_2Mn_xAl_{17-x}$ and $Tb_2Mn_xAl_{17-x}$. The shift to the itinerant Mn moment with increasing x and/or some coherence effect of the FM Mn-moment observed in $Pd_{0.75}Mn_{0.25}B_x$ would be responsible for the metallic temperature dependence.

### 3.7. Brief Summary of Mn-Based Compounds

As in the rare-earth Fe-based compounds, the interstitial atoms give rise to the enhancement of FM interaction in the weak hybridization regime leading to the appearance of room temperature ferromagnetism. However, the Mn compounds surveyed above manifest the change or additional formation of magnetism by the interstitial atoms, while many rare-earth Fe-based parent compounds are already ferromagnets. The change from paramagnetic to FM state is observed in hydrogen-absorbed $Th_6Mn_{23}$, hydrogen-absorbed $YMn_2$ or $Sm_2Mn_8Al_9B_x$. The result of $Mn_5Si_3C_x$ thin film may be a rare example of change from the AFM to FM state by the interstitial atoms. In $Pd_{0.75}Mn_{0.25}B_x$, the room temperature ferromagnetism is induced by a slight addition of boron, while the low-temperature magnetic ground state of the parent compound is unchanged. This can be regarded as an example of the additional formation of magnetism by interstitial atoms. It should be noted that, in some cases, the change or additional formation of a magnetic state seems to abruptly occur, which is valuable for future research. We note here that the magnetic structures have been divided into FM and AFM, although some compounds may show a more complicated state such as canted AFM, and spiral AFM. In the future, discussion taking into account a more microscopic mechanism of the magnetic ordering would be necessary.

We add the comment on Mn-based Heusler compounds, which show a rich variety of physical properties such as the topological Hall effect, shape-memory and so on [54–56]. While many Mn-based Heusler compounds show ferromagnetism, a Heusler compound, allowing interstitial atoms, has not been reported to our knowledge.

### 3.8. Candidate Showing a Change in Magnetism by Interstitial Atoms

Mn compounds may be a superior platform to examine the change in the magnetic state or the additional formation of new magnetic coupling by interstitial atoms. We noted that, in some cases, the change to FM state or the additional formation of the FM state would occur irrespectively of interstitial atom-induced volume change. Notwithstanding, hereafter, we continue to discuss, based on the unit cell volume, because the results of $Mn_5Si_3C_x$, $Pd_{0.75}Mn_{0.25}B_x$ and $Sm_2Mn_8Al_9B_x$ are not well analyzed systematically by comparing several compounds with the same crystal structure. In other words, the magnetic ordering temperature vs. unit cell volume plot as shown in Figure 2 could not be constructed due to the absence of well investigated other Mn compounds in these crystal structures. Therefore, at the present stage, we believe that it is still valuable to survey the magnetic structures of compounds using the unit cell volumes. In this subsection, especially bearing a change in magnetism by interstitial atoms in mind, we seek a qualifying Mn compound using the magnetic ordering temperature vs. a unit cell volume plot. We selected the crystal structures allowing the site occupation of interstitial atoms: $Ni_3Sn$, $ThMn_{12}$, $Fe_2P$, $Ni_2In$, $LiAlSi$, $TiNiSi$, $AuCu$ and $Cu_2Sb$-type structures.

#### 3.8.1. $Ni_3Sn$-Type Structure

This hexagonal structure ($P6_3/mmc$, No. 194) is recently attractive as a topological AFM substance [20] represented by $Mn_3Sn$. All $Ni_3Sn$-type compounds displayed in Figure 12a possess only one crystallographic $6h$ site for the Mn atom. As seen in Figure 12a, only the AFM state has been observed and a change to FM state would be difficult.

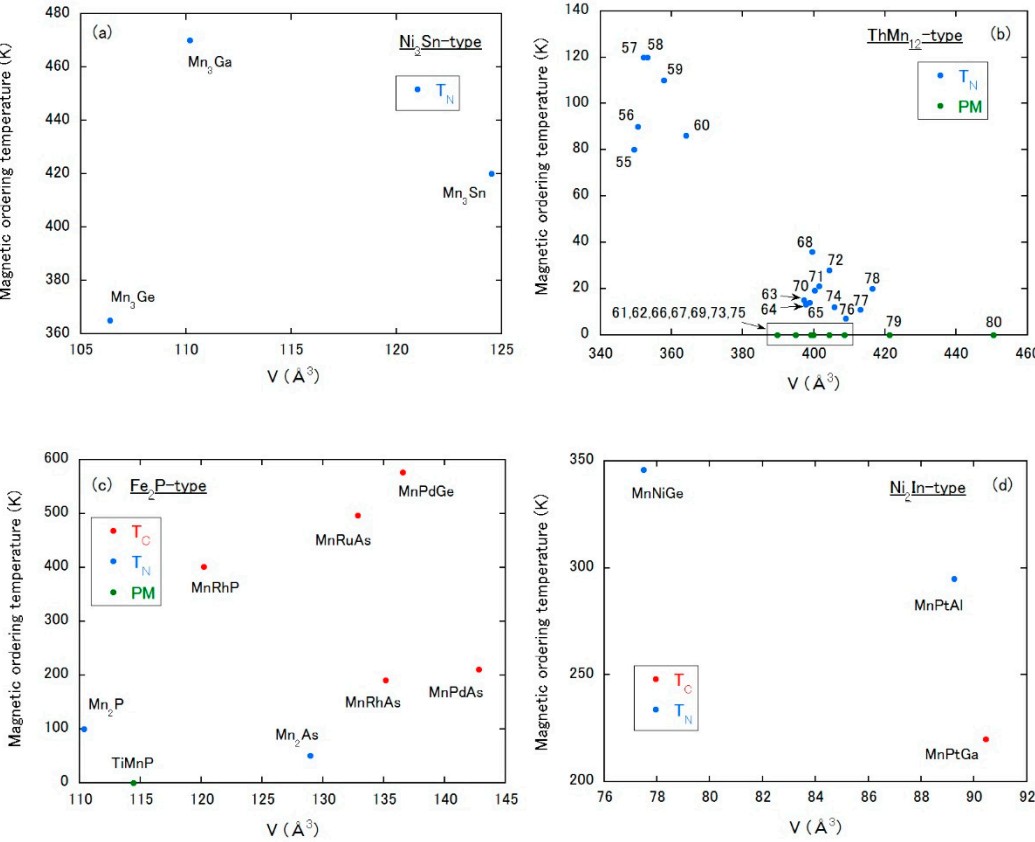

**Figure 12.** Magnetic ordering temperature vs. unit cell volume plot of Mn-based compounds with (**a**) $Ni_3Sn$-type, (**b**) $ThMn_{12}$-type, (**c**) $Fe_2P$-type and (**d**) $Ni_2In$-type, respectively. PM means a paramagnetic state down to low temperatures. The numbers in (**b**) correspond to those in Table 4.

### 3.8.2. ThMn$_{12}$-Type Structure

The tetragonal ThMn$_{12}$-type structure (*I*4/*mmm*, No. 139) is well studied in rare-earth Fe-based compounds (see Section 2.2). In the Mn compounds with this structure, Mn atoms occupy the 8*f*, 8*i*, 8*j* sites. As shown in Figure 12b, the magnetic properties are dominated by the AFM state. We note that, except for compounds No. 55–60, the AFM orderings of compounds containing rare-earth elements are triggered by the magnetic moments of rare-earth, and Mn atoms do not carry moments. Basically, with increasing volume, $T_N$ tends to be suppressed. In this class of compounds, a change from AFM to a paramagnetic state of Mn atoms by interstitial atoms may be anticipated.

### 3.8.3. Fe$_2$P-Type Structure

The compounds with the hexagonal Fe$_2$P-type structure (*P*$\bar{6}$2*m*, No. 189) displayed in Figure 12c possess the 3*g* site for the Mn atom, except Mn$_2$P and Mn$_2$As, in which there exists the 3*g* and 3*f* sites for Mn atoms. Figure 12c indicates a possible crossover from paramagnetic to FM state across the volume of approximately 115–120 Å$^3$ for compounds with only the 3*g* Mn site.

### 3.8.4. Ni$_2$In-Type Structure

The Ni$_2$In-type structure is hexagonal with the space group of *P*6$_3$/*mmc* (No. 194). Each compound has only one Mn site of the 2*a*. In this structure, $T_N$ at a smaller volume tends to be suppressed as the volume is expanded and seems to transform to $T_C$ with further increasing volume (Figure 12d). The compounds with the ThMn$_{12}$-, the Fe$_2$P- or the Ni$_2$In-type structure may be good candidates for examining a change in magnetic state by interstitial atoms as a function of the unit cell volume.

### 3.8.5. LiAlSi-Type Structure

This structure is cubic with the space group of *F*$\bar{4}$3*m* (No. 216). Each compound in Figure 13a possesses one Mn site with the 4*a*, 4*b* or 4*c*, depending on the literature. The magnetic ordering types of the cubic compounds would be classified by the VEC, which is different from the results of Figure 12a–d. The VEC values are denoted under the chemical formulae in Figure 13a. Except for the compounds with VEC = 7.7 showing the AFM state, all compounds undergo FM as one. In FM compounds, $T_C$ tends to increase as the VEC is increased. If interstitial atoms change VEC from 7.33 to 7.7, a change between FM and AFM states might be possible, which may be a new phenomenon induced by interstitial atoms.

**Table 4.** Structural and magnetic properties of Mn-based compounds. $T_{order}$ is the magnetic ordering temperature. FM, AFM and PM mean ferromagnetic, antiferromagnetic and paramagnetic, respectively. Data are from the references listed in the table.

| No. | Compound | Structure Type | *a* (Å) | *b* (Å) | *c* (Å) | *V*(Å$^3$) | $T_{order}$ (K) | Type | Ref. |
|-----|----------|----------------|---------|---------|---------|-----------|-----------------|------|------|
| 1 | Lu$_6$Mn$_{23}$ | Th$_6$Mn$_{23}$ | 12.187 | 12.187 | 12.187 | 1810.0 | 398 | FM | [57] |
| 2 | Yb$_6$Mn$_{23}$ | Th$_6$Mn$_{23}$ | 12.189 | 12.189 | 12.189 | 1810.9 | 406 | FM | [58] |
| 3 | Tm$_6$Mn$_{23}$ | Th$_6$Mn$_{23}$ | 12.226 | 12.226 | 12.226 | 1827.5 | 411 | FM | [57] |
| 4 | Er$_6$Mn$_{23}$ | Th$_6$Mn$_{23}$ | 12.275 | 12.275 | 12.275 | 1849.5 | 430 | FM | [57] |
| 5 | Ho$_6$Mn$_{23}$ | Th$_6$Mn$_{23}$ | 12.324 | 12.324 | 12.324 | 1871.8 | 445 | FM | [57] |
| 6 | Dy$_6$Mn$_{23}$ | Th$_6$Mn$_{23}$ | 12.361 | 12.361 | 12.361 | 1888.7 | 447 | FM | [57] |
| 7 | Tb$_6$Mn$_{23}$ | Th$_6$Mn$_{23}$ | 12.396 | 12.396 | 12.396 | 1904.8 | 457 | FM | [57] |
| 8 | Y$_6$Mn$_{23}$ | Th$_6$Mn$_{23}$ | 12.438 | 12.438 | 12.438 | 1924.2 | 500 | FM | [59] |
| 9 | Gd$_6$Mn$_{23}$H$_{24}$ | Th$_6$Mn$_{23}$ | 12.515 | 12.515 | 12.515 | 1960.2 | 260 | FM | [60] |
| 10 | Th$_6$Mn$_{23}$ | Th$_6$Mn$_{23}$ | 12.523 | 12.523 | 12.523 | 1963.9 | 0 | PM | [59] |
| 11 | Sm$_6$Mn$_{23}$ | Th$_6$Mn$_{23}$ | 12.558 | 12.558 | 12.558 | 1980.4 | 456 | FM | [57] |
| 12 | Gd$_6$Mn$_{23}$ | Th$_6$Mn$_{23}$ | 12.578 | 12.578 | 12.578 | 1989.9 | 489 | FM | [57] |
| 13 | Nd$_6$Mn$_{23}$ | Th$_6$Mn$_{23}$ | 12.657 | 12.657 | 12.657 | 2027.6 | 441 | FM | [57] |

Table 4. *Cont.*

| No. | Compound | Structure Type | $a$ (Å) | $b$ (Å) | $c$ (Å) | $V$ (Å$^3$) | $T_{order}$ (K) | Type | Ref. |
|---|---|---|---|---|---|---|---|---|---|
| 14 | $Y_6Mn_{23}H_{23}$ | $Th_6Mn_{23}$ | 12.805 | 12.805 | 12.805 | 2099.6 | 180 | AFM | [61] |
| 15 | $Th_6Mn_{23}D_{16}$ | $Th_6Mn_{23}$ | 12.922 | 12.922 | 12.922 | 2157.7 | 329 | FM | [62] |
| 16 | $Gd_{12}Mn_{45}H_{43}$ | $Th_6Mn_{23}$ | 12.97 | 12.97 | 12.97 | 2181.8 | 180 | FM | [63] |
| 17 | $Tb_6Mn_{23}H_{23}$ | $Th_6Mn_{23}$ | 13.017 | 13.017 | 13.017 | 2205.6 | 220 | FM | [64] |
| 18 | $Sm_6Mn_{23}H_{24}$ | $Th_6Mn_{23}$ | 13.04 | 13.04 | 13.04 | 2217.3 | 230 | FM | [60] |
| 19 | $Nd_6Mn_{23}H_{24}$ | $Th_6Mn_{23}$ | 13.237 | 13.237 | 13.237 | 2319.4 | 220 | FM | [60] |
| 20 | $Th_6Mn_{23}H_{30}$ | $Th_6Mn_{23}$ | 13.259 | 13.259 | 13.259 | 2330.9 | 329 | FM | [59] |
| 21 | $YMn_2Si_2$ | $ThCr_2Si_2$ | 3.924 | 3.924 | 10.457 | 161.0 | 463 | AFM | [65] |
| 22 | $LaMn_2Si_2$ | $ThCr_2Si_2$ | 4.1151 | 4.1151 | 10.612 | 179.7 | 305 | FM | [65] |
| 23 | $CeMn_2Si_2$ | $ThCr_2Si_2$ | 3.99 | 3.99 | 10.51 | 167.3 | 377 | AFM | [65] |
| 24 | $PrMn_2Si_2$ | $ThCr_2Si_2$ | 4.03 | 4.03 | 10.559 | 171.5 | 350 | AFM | [65] |
| 25 | $NdMn_2Si_2$ | $ThCr_2Si_2$ | 4.063 | 4.063 | 10.522 | 173.7 | 368 | AFM | [65] |
| 26 | $SmMn_2Si_2$ | $ThCr_2Si_2$ | 3.975 | 3.975 | 10.52 | 166.2 | 400 | AFM | [65] |
| 27 | $EuMn_2Si_2$ | $ThCr_2Si_2$ | 3.966 | 3.966 | 10.387 | 163.4 | 391 | AFM | [66] |
| 28 | $GdMn_2Si_2$ | $ThCr_2Si_2$ | 3.948 | 3.948 | 10.468 | 163.2 | 454 | AFM | [65] |
| 29 | $TbMn_2Si_2$ | $ThCr_2Si_2$ | 3.931 | 3.931 | 10.456 | 161.6 | 504 | AFM | [65] |
| 30 | $DyMn_2Si_2$ | $ThCr_2Si_2$ | 3.915 | 3.915 | 10.44 | 160.0 | 477 | AFM | [65] |
| 31 | $HoMn_2Si_2$ | $ThCr_2Si_2$ | 3.931 | 3.931 | 10.412 | 160.9 | 455 | AFM | [65] |
| 32 | $ErMn_2Si_2$ | $ThCr_2Si_2$ | 3.905 | 3.905 | 10.42 | 158.9 | 518 | AFM | [65] |
| 33 | $TmMn_2Si_2$ | $ThCr_2Si_2$ | 3.887 | 3.887 | 10.398 | 157.1 | 500 | AFM | [65] |
| 34 | $YbMn_2Si_2$ | $ThCr_2Si_2$ | 3.877 | 3.877 | 10.391 | 156.2 | 512 | AFM | [65] |
| 35 | $LuMn_2Si_2$ | $ThCr_2Si_2$ | 3.873 | 3.873 | 10.37 | 155.6 | 468 | AFM | [65] |
| 36 | $YMn_2Ge_2$ | $ThCr_2Si_2$ | 4.0516 | 4.0516 | 10.854 | 178.2 | 395 | AFM | [65] |
| 37 | $LaMn_2Ge_2$ | $ThCr_2Si_2$ | 4.195 | 4.195 | 11.022 | 194.0 | 309 | FM | [65] |
| 38 | $CeMn_2Ge_2$ | $ThCr_2Si_2$ | 4.144 | 4.144 | 10.97 | 188.4 | 325 | FM | [65] |
| 39 | $PrMn_2Ge_2$ | $ThCr_2Si_2$ | 4.123 | 4.123 | 10.929 | 185.8 | 336 | FM | [65] |
| 40 | $NdMn_2Ge_2$ | $ThCr_2Si_2$ | 4.1022 | 4.1022 | 10.909 | 183.6 | 334 | FM | [65] |
| 41 | $SmMn_2Ge_2$ | $ThCr_2Si_2$ | 4.062 | 4.062 | 10.896 | 179.8 | 350 | FM | [65] |
| 42 | $GdMn_2Ge_2$ | $ThCr_2Si_2$ | 4.029 | 4.029 | 10.895 | 176.9 | 368 | AFM | [65] |
| 43 | $TbMn_2Ge_2$ | $ThCr_2Si_2$ | 4.006 | 4.006 | 10.875 | 174.5 | 418 | AFM | [65] |
| 44 | $DyMn_2Ge_2$ | $ThCr_2Si_2$ | 3.989 | 3.989 | 10.863 | 172.9 | 386 | AFM | [65] |
| 45 | $HoMn_2Ge_2$ | $ThCr_2Si_2$ | 3.977 | 3.977 | 10.847 | 171.6 | 404 | AFM | [65] |
| 46 | $ErMn_2Ge_2$ | $ThCr_2Si_2$ | 3.948 | 3.948 | 10.791 | 168.2 | 395 | AFM | [65] |
| 47 | $YbMn_2Ge_2$ | $ThCr_2Si_2$ | 4.067 | 4.067 | 10.871 | 179.8 | 341 | AFM | [65] |
| 48 | $ThMn_2Si_2$ | $ThCr_2Si_2$ | 4.0225 | 4.0225 | 10.475 | 169.5 | 486 | AFM | [65] |
| 49 | $UMn_2Si_2$ | $ThCr_2Si_2$ | 3.922 | 3.922 | 10.284 | 158.2 | 377 | FM | [67] |
| 50 | $ThMn_2Ge_2$ | $ThCr_2Si_2$ | 4.084 | 4.084 | 10.93 | 182.3 | 400 | AFM | [65] |
| 51 | $UMn_2Ge_2$ | $ThCr_2Si_2$ | 4.012 | 4.012 | 10.803 | 173.9 | 390 | FM | [67] |
| 52 | $Mn_3Ge$ | $Ni_3Sn$ | 5.338 | 5.338 | 4.312 | 106.4 | 365 | AFM | [68] |
| 53 | $Mn_3Ga$ | $Ni_3Sn$ | 5.404 | 5.404 | 4.357 | 110.2 | 470 | AFM | [69] |
| 54 | $Mn_3Sn$ | $Ni_3Sn$ | 5.64 | 5.64 | 4.52 | 124.5 | 420 | AFM | [20] |
| 55 | $ErMn_{12}$ | $ThMn_{12}$ | 8.5719 | 8.5719 | 4.7553 | 349.4 | 80 | AFM | [70] |
| 56 | $HoMn_{12}$ | $ThMn_{12}$ | 8.5817 | 8.5817 | 4.7592 | 350.5 | 90 | AFM | [71] |
| 57 | $YMn_{12}$ | $ThMn_{12}$ | 8.597 | 8.597 | 4.7637 | 352.1 | 120 | AFM | [72] |
| 58 | $TbMn_{12}$ | $ThMn_{12}$ | 8.6076 | 8.6076 | 4.7666 | 353.2 | 120 | AFM | [73] |
| 59 | $DyMn_{12}$ | $ThMn_{12}$ | 8.67 | 8.67 | 4.76 | 357.8 | 110 | AFM | [74] |
| 60 | $GdMn_{12}$ | $ThMn_{12}$ | 8.673 | 8.673 | 4.839 | 364.0 | 86 | AFM | [74] |
| 61 | $ScMn_4Al_8$ | $ThMn_{12}$ | 8.7734 | 8.7734 | 5.0629 | 389.7 | 0 | PM | [75] |
| 62 | $LuMn_4Al_8$ | $ThMn_{12}$ | 8.814 | 8.814 | 5.083 | 394.9 | 0 | PM | [75] |
| 63 | $ErMn_4Al_8$ | $ThMn_{12}$ | 8.829 | 8.829 | 5.096 | 397.2 | 15 | AFM | [76] |
| 64 | $TmMn_4Al_8$ | $ThMn_{12}$ | 8.848 | 8.848 | 5.08 | 397.7 | 13 | AFM | [76] |
| 65 | $HoMn_4Al_8$ | $ThMn_{12}$ | 8.845 | 8.845 | 5.097 | 398.8 | 14 | AFM | [76] |
| 66 | $UMn_4Al_8$ | $ThMn_{12}$ | 8.8474 | 8.8474 | 5.0993 | 399.2 | 0 | PM | [77] |
| 67 | $YMn_4Al_8$ | $ThMn_{12}$ | 8.86 | 8.86 | 5.09 | 399.6 | 0 | PM | [76] |
| 68 | $GdMn_6Al_6$ | $ThMn_{12}$ | 8.845 | 8.845 | 5.108 | 399.6 | 36 | AFM | [78] |
| 69 | $YbMn_4Al_8$ | $ThMn_{12}$ | 8.854 | 8.854 | 5.102 | 400.0 | 0 | PM | [76] |

**Table 4.** *Cont.*

| No. | Compound | Structure Type | $a$ (Å) | $b$ (Å) | $c$ (Å) | $V$ (Å³) | $T_{order}$ (K) | Type | Ref. |
|-----|----------|----------------|---------|---------|---------|----------|-----------------|------|------|
| 70 | $DyMn_4Al_8$ | $ThMn_{12}$ | 8.849 | 8.849 | 5.112 | 400.3 | 19 | AFM | [76] |
| 71 | $TbMn_4Al_8$ | $ThMn_{12}$ | 8.865 | 8.865 | 5.108 | 401.4 | 21 | AFM | [76] |
| 72 | $GdMn_4Al_8$ | $ThMn_{12}$ | 8.887 | 8.887 | 5.119 | 404.3 | 28 | AFM | [76] |
| 73 | $TbMn_{4.04}Al_{7.96}$ | $ThMn_{12}$ | 8.8885 | 8.8885 | 5.1179 | 404.3 | 0 | PM | [79] |
| 74 | $SmMn_4Al_8$ | $ThMn_{12}$ | 8.902 | 8.902 | 5.12 | 405.7 | 12 | AFM | [76] |
| 75 | $CeMn_4Al_8$ | $ThMn_{12}$ | 8.89 | 8.89 | 5.17 | 408.6 | 0 | PM | [76] |
| 76 | $NdMn_4Al_8$ | $ThMn_{12}$ | 8.925 | 8.925 | 5.133 | 408.9 | 7 | AFM | [76] |
| 77 | $PrMn_4Al_8$ | $ThMn_{12}$ | 8.962 | 8.962 | 5.143 | 413.1 | 11 | AFM | [76] |
| 78 | $EuMn_4Al_8$ | $ThMn_{12}$ | 8.982 | 8.982 | 5.161 | 416.4 | 20 | AFM | [76] |
| 79 | $LaMn_4Al_8$ | $ThMn_{12}$ | 9.031 | 9.031 | 5.166 | 421.3 | 0 | PM | [76] |
| 80 | $ThMn_4Al_8$ | $ThMn_{12}$ | 8.937 | 8.937 | 5.639 | 450.4 | 0 | PM | [80] |
| 81 | $Mn_2P$ | $Fe_2P$ | 6.074 | 6.074 | 3.454 | 110.4 | 100 | AFM | [81, 82] |
| 82 | $TiMnP$ | $Fe_2P$ | 6.188 | 6.188 | 3.451 | 114.4 | 0 | PM | [83] |
| 83 | $MnRhP$ | $Fe_2P$ | 6.226 | 6.226 | 3.581 | 120.2 | 401 | FM | [84] |
| 84 | $Mn_2As$ | $Fe_2P$ | 6.3627 | 6.3627 | 3.6784 | 129.0 | 50 | AFM | [85] |
| 85 | $MnRuAs$ | $Fe_2P$ | 6.5155 | 6.5155 | 3.614 | 132.9 | 496 | FM | [84] |
| 86 | $MnRhAs$ | $Fe_2P$ | 6.482 | 6.482 | 3.714 | 135.1 | 190 | FM | [86] |
| 87 | $MnPdGe$ | $Fe_2P$ | 6.639 | 6.639 | 3.577 | 136.5 | 576 | FM | [87] |
| 88 | $MnPdAs$ | $Fe_2P$ | 6.626 | 6.626 | 3.756 | 142.8 | 210 | FM | [84] |
| 89 | $MnNiGe$ | $Ni_2In$ | 4.078 | 4.078 | 5.381 | 77.50 | 346 | AFM | [88] |
| 90 | $MnPtAl$ | $Ni_2In$ | 4.329 | 4.329 | 5.499 | 89.25 | 295 | AFM | [89] |
| 91 | $MnPtGa$ | $Ni_2In$ | 4.328 | 4.328 | 5.576 | 90.45 | 220 | FM | [90] |
| 92 | $NiMnSb$ | $LiAlSi$ | 5.928 | 5.928 | 5.928 | 208.3 | 728 | FM | [91] |
| 93 | $MnIrAl$ | $LiAlSi$ | 5.981 | 5.981 | 5.981 | 214.0 | 379 | FM | [92] |
| 94 | $MnCuSb$ | $LiAlSi$ | 6.095 | 6.095 | 6.095 | 226.4 | 55 | AFM | [93] |
| 95 | $MnRhSb$ | $LiAlSi$ | 6.142 | 6.142 | 6.142 | 231.7 | 320 | FM | [93] |
| 96 | $MnIrSn$ | $LiAlSi$ | 6.182 | 6.182 | 6.182 | 236.3 | 265 | FM | [94] |
| 97 | $MnPtSb$ | $LiAlSi$ | 6.21 | 6.21 | 6.21 | 239.5 | 585 | FM | [93] |
| 98 | $PdMnSb$ | $LiAlSi$ | 6.231 | 6.231 | 6.231 | 241.9 | 500 | FM | [95] |
| 99 | $MnPdTe$ | $LiAlSi$ | 6.2605 | 6.2605 | 6.2605 | 245.4 | 17 | AFM | [96] |
| 100 | $MnPtSn$ | $LiAlSi$ | 6.264 | 6.264 | 6.264 | 245.8 | 330 | FM | [93] |
| 101 | $MnAuSn$ | $LiAlSi$ | 6.4313 | 6.4312 | 6.4312 | 266.0 | 740 | FM | [97] |
| 102 | $MnRhSi$ | $TiNiSi$ | 6.1994 | 3.7968 | 7.1387 | 168.0 | 367 | AFM | [98] |
| 103 | $HfMnP$ | $TiNiSi$ | 6.3257 | 3.6298 | 7.409 | 170.1 | 320 | FM | [99] |
| 104 | $ZrMnP$ | $TiNiSi$ | 6.41 | 3.657 | 7.515 | 176.2 | 370 | FM | [99] |
| 105 | $LuMnSi$ | $TiNiSi$ | 6.82 | 3.962 | 7.839 | 211.8 | 255 | AFM | [100] |
| 106 | $UMnGe$ | $TiNiSi$ | 6.866 | 4.2594 | 7.3618 | 215.3 | 240 | AFM | [101] |
| 107 | $TbMnGe$ | $TiNiSi$ | 7.077 | 4.132 | 8.166 | 238.8 | 510 | AFM | [102] |
| 108 | $GdMnGe$ | $TiNiSi$ | 7.138 | 4.1698 | 8.191 | 243.8 | 490 | AFM | [103] |
| 109 | $MnNi$ | $AuCu$ | 3.69 | 3.69 | 2.609 | 35.52 | 1070 | AFM | [104] |
| 110 | $MnPt$ | $AuCu$ | 3.855 | 3.855 | 2.726 | 40.51 | 970 | AFM | [104] |
| 111 | $MnGa$ | $AuCu$ | 3.889 | 3.889 | 2.75 | 41.59 | 629 | FM | [105] |
| 112 | $MnRh$ | $AuCu$ | 3.93 | 3.93 | 2.779 | 42.92 | 0 | PM | [104] |
| 113 | $MnPd$ | $AuCu$ | 4.069 | 4.069 | 2.877 | 47.63 | 780 | AFM | [104] |
| 114 | $MnAl$ | $AuCu$ | 3.9 | 3.9 | 3.54 | 53.84 | 578 | FM | [106] |
| 115 | $MnRh_2Sb$ | $AuCu$ | 4.171 | 4.171 | 3.494 | 60.79 | 304 | FM | [107] |
| 116 | $MnAlGe$ | $Cu_2Sb$ | 3.914 | 3.914 | 5.933 | 90.89 | 518 | FM | [108] |
| 117 | $MnGaGe$ | $Cu_2Sb$ | 3.966 | 3.966 | 5.885 | 92.57 | 440 | FM | [109] |
| 118 | $MnZnSb$ | $Cu_2Sb$ | 4.173 | 4.173 | 6.233 | 108.5 | 310 | FM | [110] |
| 119 | $LiMnAs$ | $Cu_2Sb$ | 4.267 | 4.267 | 6.178 | 112.5 | 374 | AFM | [111] |
| 120 | $YMnSi$ | $Cu_2Sb$ | 3.978 | 3.978 | 7.152 | 113.2 | 282 | FM | [112] |
| 121 | $GdMnSi$ | $Cu_2Sb$ | 4.009 | 4.009 | 7.183 | 115.5 | 314 | FM | [113] |
| 122 | $NdMnSi$ | $Cu_2Sb$ | 4.087 | 4.087 | 7.245 | 121.0 | 175 | AFM | [114] |
| 123 | $CeMnSi$ | $Cu_2Sb$ | 4.13 | 4.13 | 7.279 | 124.2 | 240 | AFM | [114] |
| 124 | $CaMnSi$ | $Cu_2Sb$ | 4.1814 | 4.1814 | 7.1429 | 124.9 | 360 | AFM | [115] |

**Table 4.** *Cont.*

| No. | Compound | Structure Type | *a* (Å) | *b* (Å) | *c* (Å) | *V*(Å³) | $T_{order}$ (K) | Type | Ref. |
|-----|----------|---------|--------|--------|--------|---------|-----------------|------|------|
| 125 | PrMnGe | Cu₂Sb | 4.196 | 4.196 | 7.344 | 129.3 | 415 | AFM | [116] |
| 126 | LaMnSi | Cu₂Sb | 4.189 | 4.189 | 7.378 | 129.5 | 310 | AFM | [114] |
| 127 | CaMnGe | Cu₂Sb | 4.23 | 4.23 | 7.27 | 130.1 | 420 | AFM | [115] |
| 128 | CeMnGe | Cu₂Sb | 4.226 | 4.226 | 7.386 | 131.9 | 415 | AFM | [116] |
| 129 | LaMnGe | Cu₂Sb | 4.277 | 4.277 | 7.449 | 136.3 | 420 | AFM | [116] |
| 130 | SrMnGe | Cu₂Sb | 4.4 | 4.4 | 7.52 | 145.6 | 254 | AFM | [117] |
| 131 | CaMnSn | Cu₂Sb | 4.4839 | 4.4839 | 7.4988 | 150.8 | 250 | AFM | [117] |
| 132 | BaMnGe | Cu₂Sb | 4.5 | 4.5 | 7.92 | 160.4 | 425 | AFM | [117] |
| 133 | SrMnSn | Cu₂Sb | 4.662 | 4.662 | 7.749 | 168.4 | 252 | AFM | [117] |

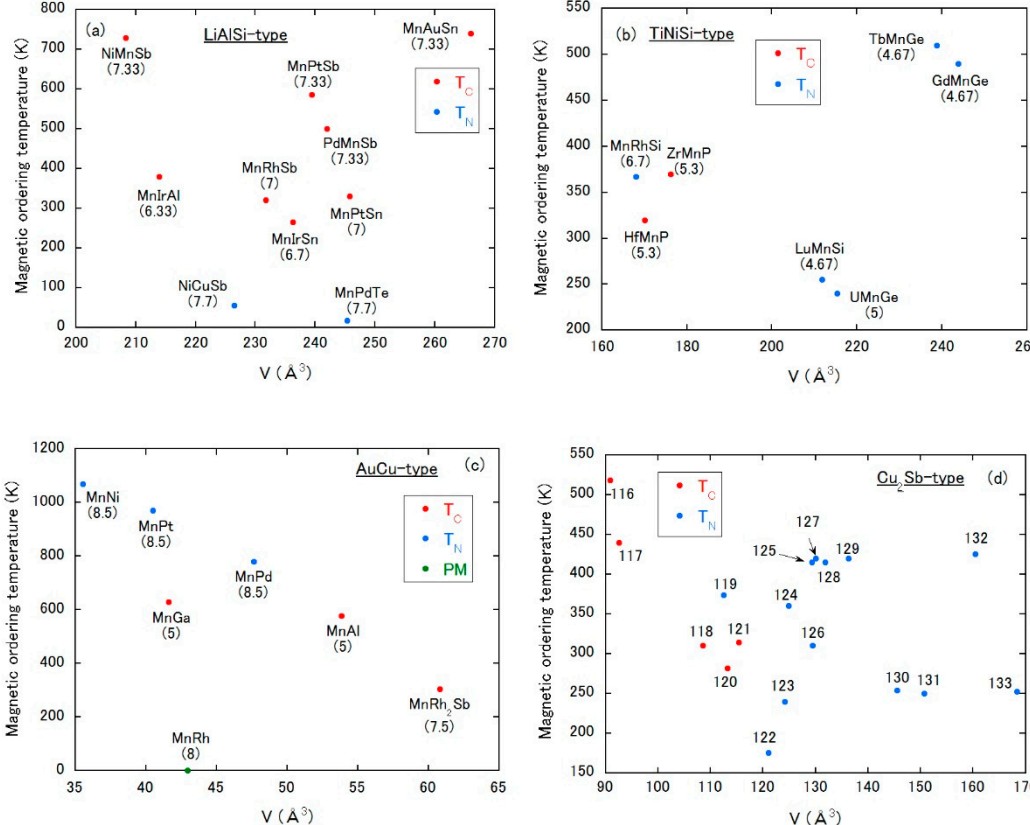

**Figure 13.** Magnetic ordering temperature vs. unit cell volume plot of Mn-based compounds with (**a**) LiAlSi-type, (**b**) TiNiSi-type, (**c**) AuCu-type and (**d**) Cu₂Sb-type, respectively. PM means a paramagnetic state down to low temperatures. In (**a**–**c**), valence electron count per atom (VEC) is denoted under each chemical formula. The numbers in (**d**) correspond to those in Table 4.

### 3.8.6. TiNiSi-Type Structure

This is the famous orthorhombic structure with the space group *Pnma* (No. 62). Mn atoms occupy the 4*c* site. As shown in Figure 13b, the border between AFM and FM states may be situated at *V* = 180–200 Å³. Moreover, only the compounds accompanied by VEC = 5.3 show the FM ground state. The interstitial alloying with the change in VEC might be effective for a magnetic state change.

### 3.8.7. AuCu-Type Structure

The AuCu-type structure is tetragonal with the space group of *P4/mmm* (No. 123). Except for MnRh₂Sb, Mn atoms occupy two crystallographic sites called the 1*a* and 1*c* sites. In MnRh₂Sb, Mn sites

are reduced to the 1*a* one. As in the LiAlSi-type structure, the magnetic ordering type would be correlated with VEC; the AFM or paramagnetic state is observed for a VEC larger than 8, and the FM state for VEC = 5 or 7.5. Under a fixed VEC value, $T_N$ systematically decreases with increasing volume, while the volume dependence of $T_C$ is very weak for ferromagnets.

### 3.8.8. $Cu_2Sb$-Type Structure

This is the tetragonal structure (*P*4/*nmm*, No. 129), in which Mn atoms occupy the 2*a* site. Contrary to the prediction of the Bethe–Slater curve, a volume expansion favors an AFM state. The magnitude of the magnetic ordering temperature is likely suppressed by expanding the volume.

## 4. Rare-Earth-Based Compounds

### 4.1. $R_5Si_3B_x$

The $Mn_5Si_3$-type $R_5Si_3$ allows for interstitial boron atoms, which leads to volume reduction with an increasing boron concentration [118]. For the parent compound with x = 0, the AFM orderings are observed in R = Gd and Tb at $T_N$ = 75 and 69 K, respectively. On the other hand, R = Dy and Ho show the FM orderings at $T_C$ = 120 and 11 K, respectively. By adding boron atoms, the $T_N$ of R = Gd and Tb are slightly reduced to 67 K in both cases, and the changes of $T_C$ in R = Dy and Ho are also subtle; $T_C$ remains at 120 K in R = Dy and is slightly enhanced to 15 K in R = Ho. In each compound, $\mu_{eff}$ does not so largely depend on the boron addition, which means a weak hybridization between rare earth and boron atoms.

### 4.2. $NdScSiC_x$

This compound crystallizes into the tetragonal $La_2Sb$-type structure (*I*4/*mmm*, No. 139) [119]. The interstitial carbon atoms expand the *a* axis and contract the *c* axis. The latter fact, in particular, enhances the chemical bonding between Nd and C and decreases $T_C$ = 171 K in the parent NdScSi to 50 K at x = 0.5. Taking into account that the 4*f* orbital of a rare-earth atom is usually well localized, the drastic change in magnetic ordering temperature is unexpected as in $R_5Si_3B_x$. Thus, the large modification of $T_C$ in this compound is very interesting.

## 5. Perspectives

### 5.1. Application of Mn-Based Magnetic Materials

The change in magnetism between FM and AFM states or the additional formation of new magnetic coupling at rather high temperatures by interstitial atoms is substantially valuable in a magnetic device integrated with both FM and AFM materials due to the easy on-demand control of magnetism in the fabrication process. At the present stage, Mn-based compounds fulfill the requirement of change or the additional formation of magnetism, while in Fe-based compounds, only improvements of FM properties are extensively investigated and the change (or the additional formation) of the magnetic state is not well explored. Focusing on the research area of the permanent magnet, a rare-earth Mn-based permanent magnet is still missing, although the MnBi-type magnets are well known. Based on the Bethe–Slater curve, Mn atoms favor the FM state with expanding Mn–Mn distance. Therefore, the density of Mn could not be so increased as in rare-earth Fe-based permanent magnets, resulting in a smaller saturation magnetization. However, there exists a large gap of *BH* energy product between the NdFeB magnets and ferrite magnets, and a rare-earth Mn-based permanent magnet may be a good candidate filling the gap [120].

### 5.2. Towards Further on-Demand Control of Magnetism

Further improvement of magnetic properties in the on-demand control would be achieved by another strategy such as a composition effect and carrier doping. If a metallurgical phase diagram of

a target compound possesses a homogeneity range, the magnetic ordering temperature often varies with the atomic composition. For example, the $T_C$ of $Tb_2Co_2Ga$ ranges rather widely from 75 to 145 K by changing the starting composition [121]. Such a composition effect is reported in other compounds [122–124] such as $Nd_3Pd_{20}Ge_6$, $Tb_3Co_3Ga$ and $Mn_{1+x}Ga$. In many cases, the crystal structure parameters slightly change, which heavily affects the magnetic exchange interactions.

The magnetic anisotropy energy is one of the important factors in characterizing a ferromagnet. It is known that it can be tuned by doping, mainly due to the variation of the density of states near the Fermi level. The doping effect is reported in, for example, $Ni_2MnGa$, $SmCo_{5-x}Fe$, MnBi, $Nd_2Fe_{17}X$ (X = C or N), $Ce_2AuP_3$ and so on [125–129]. Taking into account the crystal symmetry, which is related to the magnetic anisotropy energy, a lower crystal symmetry with more tunable crystal parameters might be favorable.

### 5.3. Comments on Control of Magnetism by External Field

Another interesting control of magnetism is the manipulation of spin by external fields such as the electric field and the optical light. For example, the employment of an electric double-layer transistor has achieved a control of magnetism by weak voltage [130]. Recently, an optical change in magnetism through the Kondo effect has been reported [131]. The interstitial atoms may precisely control the magnetism so that only a small magnitude of external field is required for the device working, and the power consumption can be highly reduced.

### 5.4. Comments on Critical Behavior

From the fundamental viewpoint, research into critical behavior is interesting. Actually, in strongly correlated electron systems, there have been plenty of studies for seeking a quantum critical point under the suppression of magnetism [132–136]. We speculate that a formation of FM exchange coupling above room temperature would be a discontinuous phenomenon as mentioned in the results of Mn-based compounds. While it is not well investigated for rare-earth Fe-based compounds, we note that $RFe_{11}TiC_x$ and $RFe_{11}TiC_x$ show a finite change in $T_C$ at an infinitely zero value of volume expansion depending on R species [137].

## 6. Summary

This review surveyed the studies of interstitial atoms in rare-earth Fe-, Mn-, rare-earth-based magnetic materials, especially focusing on the Mn-based compounds since the effect of interstitial atoms is not well investigated. The light elements would occupy the interstitial sites in the respective crystal structure, and change the unit cell volume, although the degree of change depends on the strength of orbital hybridization between the magnetic and interstitial atoms. The light elements are occasionally essential for the stabilization of the desired crystal structure. In the abundant studies of rare-earth Fe-based permanent magnets, the role of interstitial atoms seems to be restricted to the enhancements of $T_C$ and saturated magnetization, and the change in easy magnetization direction. In several Mn-based compounds, magnetic ordering temperature and the magnetization also tends to be increased with the increasing density of interstitial atoms after the appearance of FM states. However, it is peculiar for the Mn-based compounds that the change or additional formation of magnetism by interstitial atoms is possible: the change from the AFM (paramagnetic) to FM state or the additional formation of the FM state coexisting with the ground state of Mn atoms in the parent compound. Furthermore, the FM exchange coupling would abruptly emerge under a slight addition of interstitial elements, and this is an important research topic for a deep understanding of interstitial atom engineering. The candidates of Mn-based compounds, possibly showing the change in magnetism by interstitial atoms, are briefly discussed. We note that not only the unit cell volume but also the VEC should be taken into account to design the change in magnetism. The change between AFM and FM states by just controlling the number of interstitial atoms is a very promising elemental technology for making highly functional magnetic devices integrated with both FM and AFM materials without a large lattice mismatch.

**Author Contributions:** Conceptualization, J.K., N.S. and M.T.; methodology, J.K., N.S. and M.T.; formal analysis, J.K., K.S., T.H. and F.H.; writing—original draft preparation, J.K.; writing—review and editing, J.K., N.S. and M.T. All authors have read and agreed to the published version of the manuscript.

**Funding:** This research received no external funding.

**Acknowledgments:** J.K. is grateful for the support provided by Comprehensive Research Organization of Fukuoka Institute of Technology.

**Conflicts of Interest:** The authors declare no conflict of interest.

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
