# Peer review of "Interstitial Atom Engineering in Magnetic Materials"

_metals, doi:10.3390/met10121644_

Round 1

Reviewer 1 Report

  1. Authors collect and analyze plenty of experimental result data laboriously to display various compound systems' magnetic properties, which makes this paper worth reading. Although some results could not have consistent explanations of magnetic mechanisms, this paper is still a useful reference for researchers engaging in this research field.
  2. This paper's main drawback is only dividing the magnetic order of the magnetic compounds into two categories, i.e., AFM and FM. The magnetic order is more complicated than pure FM and AFM, such as canted AFM order, spiral AFM,…all of them corresponding to individual magnetic formation mechanisms. The authors also intend to conduct the magnetic properties by the interstitial light atoms through the volume change or the orbital hybridization change, ignoring the more microscopic mechanism of the magnetism formation. In short, to simple to describe the magnetic properties of the magnetic materials.

Author Response

Thank you for your valuable comments. We respectfully respond to your comments.

  • Comment: Authors collect and analyze plenty of experimental result data laboriously to display various compound systems' magnetic properties, which makes this paper worth reading. Although some results could not have consistent explanations of magnetic mechanisms, this paper is still a useful reference for researchers engaging in this research field.

        Response: We appreciate your encouraging comment.

  • Suggestion: This paper's main drawback is only dividing the magnetic order of the magnetic compounds into two categories, i.e., AFM and FM. The magnetic order is more complicated than pure FM and AFM, such as canted AFM order, spiral AFM,…all of them corresponding to individual magnetic formation mechanisms. The authors also intend to conduct the magnetic properties by the interstitial light atoms through the volume change or the orbital hybridization change, ignoring the more microscopic mechanism of the magnetism formation. In short, to simple to describe the magnetic properties of the magnetic materials.

        Response: Following the comment, we have added the note in line 365-368.           In this note, we have pointed out that discussion taking into account more             microscopic mechanism of the magnetic ordering is necessary in the future.

Reviewer 2 Report

Review of the manuscript: Interstitial atom engineering in magnetic materials (metals-1023892-peer-review-v1)

In this review paper is reported almost all main research on interstitial atoms in Mn-based compounds. Some reports on interstitial atoms in Rare-earth Fe-based compounds (like R2Fe17Nx were R=light rare-earth, ThMn12-type and BaCd11-type) are also mentioned.

The information comes from a reliable, trusted source such as academic journals or academic books. The structure and flow are in good standing.

The manuscript is expected to say something different and indicates the best avenues for future research, giving reader information that he couldn't find elsewhere. In this view, we can consider this attempt was  successfully done.

An overall evaluation of the manuscript is attached.

Author Response

Thank you for your valuable comments. We respectfully respond to your comments.

  • Response for Authority and Reliability: We appreciate your encouraging comment.
  • Balance

        (2-1) Comment: The authors had mentioned old reports on MnBi magnets          [ref 13-14], while remarkable reports on MnBi modified magnets published            recently are not cited. What is more, in ref [13-14] was not reported any              product final of magnets. Reports also on carbon-doped Mn-Al alloys might be        mentioned.

        Response: We have added ref [16-18], which are recently published and              contain the reports of magnetic properties of MnBi magnets and carbon-                doped Mn-Al alloys.

       (2-2) Comment: Taking in consideration the missed information for some              class of interstitials, the statement: ''The systematic information of effect              atoms in rare earth Fe… -" is not applied here. Such interstitial are                        R3(FeM)29X (C=N, C, H) or other similar intermetallic compounds, dopants          on  the Fe3Sn phase (beside mentioned Mn3Sn etc), as well as review on last        research supported by European Commision, USA etc (like Novamag etc),              which would complete the so-called statement "systematic".

       Response: We have deleted “systematic” in line 71, because our main                    purpose is the survey of Mn-based compounds. However, we have added              recent reviews including by Novamag as ref [3-6] to suggest that interstitial          compounds are enough current interest for readers.

      (2-3) Comment: Another principle is that the authors need to avoid a review         that relies on self- citations. There are 11 self-citations out of 111 in total (8.7       %), some of which are not relevant to this review like ref [44].

      Response: Following the comment, we have deleted ref [44].

  • Response for Timeliness: We have added 15 references [3-6], [16-18], [35-39] and [54-56], in which 13 references are from 2-5 years, in order to improve the statistic. Following the comment, we have enlarged the topic of interstitial R(Fe,M)12 compounds in line 119-126.
  • Response for Novelty: We appreciate your encouraging comment.
  • Scientific accuracy

       (5-1) Comment: A large class of Mn-based alloys are the Heusler magnetic           alloys. Only three relatively old references are given on these type of alloys           [83, 86, 96]. In the last decade there is a large research on Heusler alloys. It         would be of interest if authors would have considered new reports on Heusler         alloys, giving an opinion of the possible interstitials Heusler alloys. At least              one such review was report on Progress in Solid State Chemistry.

       Response: We have added the comment on Mn-based Heusler compounds in         line 369-372 with the opinion of the possible interstitials Heusler alloys. We           also added recent references [54-56] of Mn-based Heusler compounds.

       (5-2) Comment: In the reference [31], line 111, is reported the maximum             energy product of the powders, not any magnet with such value of BH was           produced, as incorrectly someone would understand.   

       Response: Following the comment, we have corrected the sentences in line           143-145.

       (5-3) Comment: In Line 408 is this statement: “…a rare-earth Mn-based               permanent magnet is still missing”. This statement is not quite exact. There           are many recent reports on successful synthesizes of MnBi-type magnets not         mention in this review. Mentioning ref. [116], which deals with theoretical             calculations, does not fill this gap.   

       Response: We have added the brief comment on MnBi-type magnets in line           733-734. To avoid the misunderstanding of readers, we have slightly                     changed the sentence in line 738.

       (5-4) Comment: Some references, although from the past 2–5 years, are             irrelevant to this review paper (like ref [3-4-5] etc).    

       Response: We have deleted ref. [3-5].